# Reinforced Structure Effect on Thermo-Oxidative Stability of Polymer-Matrix Composites: 2-D Plain Woven Composites and 2.5-D Angle-Interlock Woven Composites

**DOI:** 10.3390/polym14173454

**Published:** 2022-08-24

**Authors:** Xingzhong Gao, Tiancong Han, Bolin Tang, Jie Yi, Miao Cao

**Affiliations:** 1School of Textile Science and Engineering, Xi’an Polytechnic University, Xi’an 710048, China; 2Key Laboratory of Functional Textile Material and Product, Xi’an Polytechnic University, Ministry of Education, Xi’an 710048, China; 3Key Laboratory of Yarn Materials Forming and Composite Processing Technology of Zhejiang Province, College of Material and Textile Engineering, Jiaxing University, Jiaxing 314001, China

**Keywords:** thermo-oxidative stability, woven composites, structure effect, finite element

## Abstract

The thermo-oxidative stability of carbon fiber polymer matrix composites with different integral reinforced structures was investigated experimentally and numerically. Specimens of 2-D plain woven composites and 2.5-D angle-interlock woven composites were isothermally aged at 180 °C in hot air for various durations up to 32 days. The thermal oxidative ageing led to the degradation of the matrix and the fiber/matrix interface. The degradation mechanisms of the matrix were examined by ATR-FTIR and thermal analysis. The interface cracks caused by thermal oxidative ageing were sensitive to the reinforced structure. The thermo-oxidative stability of the two composites was numerically compared in terms of matrix shrinking and crack evolution and then experimentally validated by interlaminar shear tests.

## 1. Introduction

Advanced textile composites are gaining market share in various industries, including examples in the aerospace [1], maritime [2], automotive [3], civil infrastructure [4,5], and wearable electronics [6] industries, due to their exceptional electrical, mechanical, and thermal properties. Woven composites are becoming a research hotspot as they are one of the most advanced textile composites [7,8,9] and have shown great potential for aircraft applications, such as wings and engine blades, due to the high strength/weight ratio and impact resistance [10,11,12]. Aerospace applications require a long service life of materials while the thermal oxygen ambient will always be met during the service, which causes the reduction of composite properties and threatens aircraft safety [13].

The polymer matrix is susceptible to temperatures. The thermal properties of the polymer matrix have been widely reported by previous work [14,15] using dynamic mechanical analysis, thermogravimetric analysis, differential scanning calorimeter analysis, and so on. The obtained glass transition temperature or decomposition temperature could be regarded as an important indicator for thermal stability. The elevated temperature can also accelerate the oxidation rate, and the combined effect of thermolysis and oxidation promotes chain scission, accompanied by the departure of low-molecule volatiles, and finally leads to chemical shrinkage [16,17,18]. The matrix shrinkage has been quantitatively measured at a microscopic scale by many scholars [19,20,21,22]. While in the fiber reinforced polymer composites, most of the reinforcement, such as carbon fiber, is reasonably inert to thermal oxidation in service (<250 °C) [23]. Thus, the matrix shrinkage causes a mismatch of deformation between the fiber and matrix and then induces a tensile stress within the fiber–matrix interfacial phase [24]. The stress accumulates with ageing time and eventually leads to interface cracks, which create additional pathways for oxygen diffusion and compromise the structural integrity of composite materials [25,26].

The distribution of interface cracks shows the relationship of the architectural structure of fibers. For composite laminates, the ply orientation angles [27], fiber spacing [28,29] as well as stacking sequences [30] have a significant influence on the initiation and propagation of the cracks. The evolution of ageing cracks is much harder to predict in fabric reinforced composites due to the complex interlacing structures. Anisotropic distribution of ageing cracks has been reported in braided and woven composites due to different yarn spatial configurations [31,32,33,34]. The thermal oxidative stability of the composites could be affected by the reinforced structure accordingly. Wu [35] and Fan [36] compared the ageing properties of braided composites with unidirectional/laminated composites, respectively, and both results indicated that the braided composites could improve durability thanks to better structural integrity. Revealing the effect of reinforced structure on the thermo-oxidative stability of composites has good benefits for the durability design of the composite structure.

Most of the previous research has mainly focused on the ageing behaviors of composites with some specific reinforced structures. Only a few papers investigated the reinforced structure effect by comparing the degradation of overall mechanical properties. How the reinforced structure affects the distribution of matrix shrinkage and interface cracks has not yet been revealed. Herein, we present a comprehensive investigation into the thermal-oxidative degradation mechanism of 2-D plain woven composites (2-D PWC) and 2.5-D angle-interlock woven composites (2.5-D AWC). The thermal degradation mechanisms of the matrix were evaluated by chemical and thermal analysis. The reinforced structure effect on the thermal-oxidative stability of the two woven composites was explored by both numerical and experimental approaches. The structure effect was first compared by the distribution of shrinkage displacement as well as the evolution of interface cracks, and then validated by interlaminar shear tests.

## 2. Experimental

### 2.1. Material

In this paper, a diglycidyl ether of bisphenol-A (DGEBA) epoxy resin (JC-02A/JC-02B, Changshu Jaffa Chemical Inc., Suzhou, China) was selected as the resin matrix. The carbon fiber tows were provided by Toray Industries, Inc., Japan. The specifications of the epoxy matrix and carbon fiber were listed in Table 1 and Table 2, respectively.

### 2.2. Material Preparation

Figure 1 shows the structures of 2-D plain woven fabric and 2.5-D angle-interlock woven fabric. The specifications of the preforms are listed in Table 3. Epoxy resin was used to impregnate the fabric with the vacuum assisted resin transfer method (VARTM). The composite was consolidated with the curing process: 90 °C for 2 h, 110 °C for 1 h, and 130 °C for 4 h in sequence, and the vacuum was about 0.1 MPa. As the single plain woven fabric ply is very thin, the 2-D plain woven composite was obtained by stacking 20 plies of fabrics along the thickness direction (in a 0° direction). The fiber volume fraction is 44.6% and 45.2% for 2.5-D angle-interlock woven composite and 2-D plain woven composite, respectively, obtained by muffle furnace combustion.

### 2.3. Accelerated Ageing and Characterization

#### 2.3.1. Isothermal Ageing

The specimens were pretreated in an oven for 1 h at 80 °C and then divided into five groups. An unaged group (blank control group) and the other four groups were isothermally aged for 4, 8, 16, and 32 days at 180 °C in an air-circulating oven. Before ageing, all specimens were dried in the oven at 80 °C for 1 h. After ageing for a given time, the specimens were removed and cooled down to room temperature, then put into sealed bags to avoid moisture absorption.

#### 2.3.2. Chemical Analysis

Attenuated total reflectance Fourier transform infrared spectroscopy (ATR-FTIR) analyses (Nicolet 6700 FTIR spectrometer, ThermoFisher, Waltham, MA, USA) were used to determine the functional characteristics of neat resin in the surface layer before and after ageing for 32 days.

#### 2.3.3. Dynamic Mechanical Analysis (DMA)

DMA (Q800, TA Instruments, New Castle, DE, USA) was performed on resin casting in single cantilever mode with a frequency of 1 Hz over the temperature range 30 °C to 180 °C. The temperature ramping rate was 5 °C/min.

#### 2.3.4. Thermogravimetric Analysis (TGA)

Thermogravimetric (TG) studies were carried out with a TA-Instrument (TGA 4000, PerkinElmer, Waltham, MA, USA). Powder samples of about 3 mg were heated under a nitrogen atmosphere. The samples were heated from 30 to 600 °C with a ramp rate of 20 °C/min.

#### 2.3.5. Mechanical Test

The interlaminar shear tests of the specimens were conducted by the Instron universal testing machine (Instron 5967, Instron, Canton, MA, USA) at a test speed of 1 mm/min following ASTM D2344. The width-to-thickness ratio of the specimen is 2, while the span-to-thickness ratio is 4.

## 3. Numerical Analysis

### 3.1. Geometry Model

The mesoscale geometry models for both 2-D PWC (Figure 2a) and 2.5-D AWC (Figure 2b) were established based on measured structural parameters. The unit-cell of 2-D PWC consists of three fabric plies with random offset on account of the realistic staggered feature. The two models have a very close yarn volume fraction, which is 62.71% and 61.84% for PWC and AWC, respectively.

Linear tetrahedral elements (C3D4) were chosen to generate the mesh of the yarns and matrix for both PWC (Figure 3a) and AWC (Figure 3b) models. Zero-thickness cohesive elements (COH3D6) were generated to represent the interface between the yarns and the matrix. The average mesh size of the model was set to be 0.2 mm.

### 3.2. Constitutive Model

#### 3.2.1. Epoxy Resin

The epoxy resin is treated as an elastic-plastic solid obeying the J2-isotropic hardening plasticity theory with an associated flow rule and a von Mises yield criterion. The properties of the epoxy matrix were experimentally characterized by a previous study [37] and PWC, respectively.

#### 3.2.2. Yarns

Woven yarns impregnated with the epoxy resin were regarded as transversely isotropic unidirectional composite lamina. The compliance matrix of the yarns was obtained by the bridging model [38].
(1)$$\left[S\right]=\left(V_{\mathrm{fy}}\left[S^{f}\right]+V_{\mathrm{my}}\left[S^{m}\right]\left[A\right]\right){\left(V_{\mathrm{fy}}\left[I\right]+V_{\mathrm{my}}\left[A\right]\right)}^{-1}$$
where $\left[S^{f}\right]$ and $\left[S^{m}\right]$ refer to the compliance matrices of the fiber and resin. $V_{\mathrm{fy}}$ and $V_{\mathrm{my}}$ are the volume fractions of the fiber and resin matrix in yarns, respectively. [A] is a bridging matrix and [I] is a unit matrix. A fiber packing fraction $V_{\mathrm{fy}}$ of 72% was determined by dividing the realistic fiber volume fraction of the specimen by the yarn volume fraction. Table 4 lists the basic properties of constituents and calculated engineering constants of the yarns.

#### 3.2.3. Interface

The yarn-matrix interfacial properties were described by a bilinear traction-separation constitutive model (Figure 4), relating the traction (t) and separation displacement (δ) between two adjacent faces:(2)$$t=k_{p}$$
where t is the nominal traction stress vector, consisting of three components: $t_{n}$, $t_{s}$, and $t_{t}$, which represent one normal and two in-plane shear tractions respectively. $k_{p}$ is the penalty stiffness and δ is the vector of separations.

Damage initiation is predicted by quadratic nominal stress criterion:(3)$${\left\{\frac{t_{n}}{t_{n}^{0}}\right\}}^{2}+{\left\{\frac{t_{s}}{t_{s}^{0}}\right\}}^{2}+{\left\{\frac{t_{t}}{t_{t}^{0}}\right\}}^{2}=1$$

The Benzeggagh–Kenane (BK) fracture criterion [39] is adopted to control the failure evolution process. The interfacial stiffness degrades accordingly based on the damage variable D varied between 0 (undamaged interface case) and 1 (complete decohesion case):(4)$$D=\frac{\delta_{m}^{f}\left(\delta_{m}^{max}-\delta_{m}^{0}\right)}{\delta_{m}^{max}\left(\delta_{m}^{f}-\delta_{m}^{0}\right)}$$
where $\delta_{m}^{0}$, $\delta_{m}^{f}$ refers to effective separation at the initiation of damage and complete failure, respectively, and $\delta_{m}^{max}$ refers to the maximum value of the effective separation attained during the loading history. If unloading occurs before the complete decohesion, the penalty stiffness will become $k_{p}^{\prime }$, where $k_{p}^{\prime }=\left(1-D\right)k_{p}$.

Table 5 listed the interfacial parameters for the model.

##### 3.3. Modeling of Matrix Shrinkage

Matrix shrinkage leads to interface cracks, which are regarded as one of the primary thermal ageing damage modes in composites. The distribution of the shrinkage displacement field and the residual stress are closely related to the architecture of the composites. In this paper, the architecture effect was compared with the initiation and propagation of interface cracks in 2-D PWC and 2.5-D AWC. As the oxidation reaction process in terms of molecular dynamics is hard to reproduce using the finite element software ABAQUS, we employed the “shrinkage equivalent temperature difference method” [21,32] to simplify the shrinkage process of the matrix. The shrinkage displacement was reproduced by the temperature difference, which could cause the same deformation. The equivalent temperature difference can be defined by:(5)$$\Delta T_{sh}\left(t\right)=\frac{\varepsilon_{sh}\left(t\right)}{\alpha_{m}}$$
where $\varepsilon_{sh}\left(t\right)$ refers to matrix shrinkage strain at different ageing days; $\alpha_{m}$ is the thermal expansion coefficient of the epoxy matrix; and $\Delta T_{sh}\left(t\right)$ is the equivalent temperature difference.

The final resin shrinkage displacement was obtained from previous work [32] using the same material system. The temperature field load was applied to the elements within the oxidized layer using ABAQUS/Standard analysis. The boundary conditions are shown in Figure 5.

## 4. Results and Discussion

### 4.1. Thermal Oxidative Degradation Mechanisms of Epoxy Resin

#### 4.1.1. ATR-FTIR Analyses

The FTIR spectra of neat resin obtained before and after ageing for 32 days at 180 °C are shown in Figure 6. Compared with the unaged sample, several distinct changes take place in the spectrum of the unaged resin: The characteristic band of C-H near 2800~3000 cm^−1^ decreases in intensity. These phenomena demonstrate that C-H bonds were oxidized. Moreover, the characteristic absorption bands of the benzene ring near 1509 cm^−1^ (Stretching vibration of -C = C- skeleton in the benzene ring) and 828 cm^−1^ (in-plane deformation vibration of phenyl-H) decreased after 32 days of ageing, which shows that the benzene ring structure was partly destroyed. The decrease of the absorption band near 1181 cm^−1^, which was derived from the bridge between the benzene rings [40], indicates that the backbone chains of epoxy resin can be cut by thermal aging. The bands at 1235 cm^−1^ attribute to aromatic ether increases after ageing. It can be inferred that aromatic ether is formed during the ageing process as an oxidation product [41].

#### 4.1.2. Dynamic Thermomechanical Behaviors

Figure 7a shows the storage modulus over temperature after ageing at 180 °C. The storage modulus decreases gradually with the increase of ageing time in the glassy state due to chain scission. While in the rubbery state, the situation is reversed. The formation of the oxidized layer may account for this result. As mentioned above, chain scission occurs in the bridge between the benzene rings in DGEBA. Some small liberated segments escape from the system and molecular rearrangements may occur among the remainders, forming new compounds with a higher concentration of benzene ring, which have a higher glass transition temperature [16].

Figure 7b shows the loss factor (tan) over temperature. In this paper, the temperature at the peak value of tan δ is treated as the T_g_ (glass transition temperature) of the sample. At the initial stage of ageing (0–8 d), the T_g_ increases from 135 °C to 140 °C due to structural changes, such as further crosslinking and the loss of dangling chains, occurring slowly during the stage prior to the onset of severe degradation [42]. The structural changes also decrease the damping ability of the material, as indicated by the drop in the maximum tan δ values [43]. After 8 days, the ageing is dominated by thermolysis and the Tg decreases accordingly.

#### 4.1.3. Thermogravimetric Analysis

As the “skin-core” structure forms after thermo-oxidative ageing [44], the samples cut from the “skin” (i.e., the oxidized layer) and the inner core were tested, respectively (Figure 8). The temperature with 10% weight loss, T_10%_, is regarded as the initial decomposition temperature. The initial decomposition temperatures for the unaged sample and the oxidized layer and inner core are 383 °C, 374 °C, and 388 °C, respectively. Compared with the unaged one, the oxidized layer has a lower initial decomposition temperature, while the inner core’s is higher. During the ageing process, some chain scission occurs in the sample by thermolysis and then the liberated segments migrate towards the surface of the sample [17]. Some liberated segments have poorer thermal stability, which leads to the advance of the initial decomposition temperature of the oxide layer. Meanwhile, the migration of small molecules towards the surface leads to the backward decomposition of the initial temperature at the inner core. Additionally, the weight remaining in the oxidized layer was about 16% at a temperature of 600 °C, which is much higher than that of the unaged sample. This indicates the formation of a more stable compound in the oxidized layer.

### 4.2. Reinforced Structure Effect

The reinforcement architecture obviously affects the shrinkage displacement of the composite and further influences the distribution of ageing cracks [36]. In this section, the distributions of the matrix shrinkage and interface cracks in PWC and AWC were compared to reveal the effect of the reinforced structure on the thermal oxidative stability of the composites.

#### 4.2.1. Matrix Shrinkage

Figure 9 compares the local shrinkage of the matrix in PWC and AWC after thermal oxidative ageing for 16 days. The maximum calculated shrinkage depth was around 25 μm, which occurred in the resin rich zone of the top and bottom sides in both PWC and AWC.

As the shrinkage occurred, the boundary of the matrix still stuck to the yarn and internal stress was induced accordingly. The shrinkage displacement and stress induced by shrinkage were compared in two local areas between AWC and PWC (Figure 10). The AWC sample showed larger shrinkage displacement as well as internal stress compared to that of the PWC, which could be attributed to the different arrangement of the yarns. Taking six cross-sections from the surface to the interior (Figure 11), the yarn volume fraction on each surface varied from 45.4% to 77.8% in the PWC, and the fraction was kept at the constant value of 62.2% in the AWC. Meanwhile, the AWC has a larger yarn to yarn space than that of PWC, and the maximum shrinkage depth increased as the yarn to yarn spacing increased (Figure 12), as the extent of the matrix rich zone increased [29].

#### 4.2.2. Interface Crack Evolution

Figure 13 shows that the evolution of interface cracks as ageing time increased in both AWC and PWC. The cracks were initiated near the ends of weft yarns and propagated along the warp yarns as the shrinkage-induced internal stress accumulated. With the increase in ageing time, the interface cracks propagated and linked together in the PWC, forming continuous interlaminar cracks. The cracks are prone to extending under the external loading, causing delamination of the PWC specimen [45,46,47].

The cracks in the AWC were relatively discrete due to the existence of reinforced yarns toward the thickness direction, which restrained the connection of the cracks (Figure 14). Thus, the interlocked yarns along the thickness have a positive effect in improving the interlaminar properties of the AWC after thermal oxidative ageing.

### 4.3. Experimental Validates-Interlaminar Performance

Figure 15a displays the typical load–displacement curves of AWC. The curves showed linear elastic in the initial stage before the first drop occurs after the peak load. After that, the load dropped in a “zigzag” manner due to the stress redistribution caused by local damage. In contrast to AWC, the curves of PWC (Figure 15b) showed a sudden drop in load after the peak load as failure took place catastrophically at the peak stress in a brittle manner. However, the brittleness decreased with the increase of the ageing time as indicated by the curves.

Figure 16 compares the retention rate of the short beam strength along with ageing time. In the first eight days, the retention rate showed a slight declining tendency for both specimens, and the retention rate of the PWC was a bit higher than that of the AWC. While, with the increase of ageing time, the retention rate of PWC decreased significantly and became lower than that of the AWC after ageing for 16 days.

The obvious decline could be attributed to the change of failure modes after long-term exposure, as depicted in Figure 17.

The failure modes of the AWC did not change much before and after ageing; the upper side of the specimen was under compression, leading to the initiation of interface cracks. The bottom side was under tension, and the yarn’s breakage under the tension led to the final failure of the specimen.

The PWC shows different failure modes after long-term exposure. At the early stage of ageing (≤8 d), the failure is dominated by the fiber breakage at the bottom. At this stage, the PWC has good interface adhesion. The fiber tension failure occurred before the delamination failure. At this time, the short beam strength was determined by the tensile properties of the yarn, which cannot reflect the real interlaminar properties. The strength retention rate showed a slight decline consequently. With the increase of ageing time (≥16 d), the interlaminar cracks propagated, forming continuous interlaminar cracks. The cracks were prone to extending under the shear loading, causing the delamination failure of the specimen.

## 5. Conclusions

This paper investigates the thermo-oxidative stability of 2-D plain woven composites and 2.5-D angle-interlock woven composites using both experimental and numerical approaches. The specimens have been exposed to 180 °C in an air-circulating oven for 4, 8, 16, and 32 days. The combined effect of matrix degradation and interface cracks leads to the reduction of properties in polymer composites after ageing.

ATR-FTIR spectroscopy and thermal analysis were conducted to illustrate the degradation of the resin matrix during the ageing process. The combined effect of thermolysis and oxidation led to chain scission and molecular rearrangement. The newly formed oxidized layer had higher thermal stability compared to the inner core. The glass transition temperature of epoxy resin decreased due to thermolysis after long-term exposure.

The distribution of interface cracks is closely related to the reinforced structure of the composites, and a mesoscale finite element method has been established to illustrate the structure effect in terms of matrix shrinkage and crack evolution. The AWC sample showed larger shrinkage displacement as well as internal stress compared to that of the PWC due to its regular arrangement and a larger yarn to yarn space. Continuous interlaminar cracks were restrained in the AWC sample due to the existence of reinforced yarns toward the thickness direction, which had a positive effect on the improvement of the interlaminar properties of the material after thermal oxidative ageing.

Finally, the stability of interlaminar properties was experimentally estimated by short beam shear tests. The retention rate of interlaminar shear strength was 81.1% and 74.9% for AWC and PWC, respectively, after ageing for 32 days, and the results prove that the 2.5-D angle-interlock woven composites have better thermo-oxidative stability after long-term thermal exposure due to better structural integrity.

## Figures and Tables

**Figure 1 polymers-14-03454-f001:**
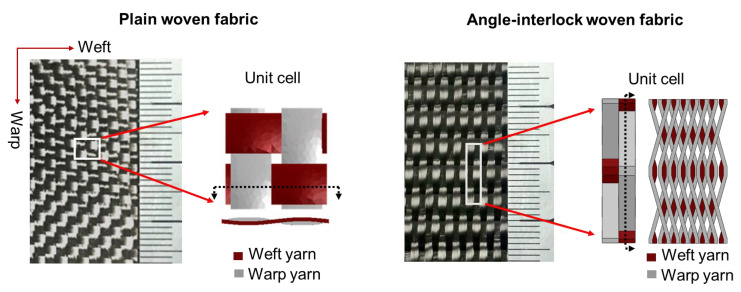
Structure of 2-D plain woven fabric and 2.5-D angle-interlock woven fabric.

**Figure 2 polymers-14-03454-f002:**
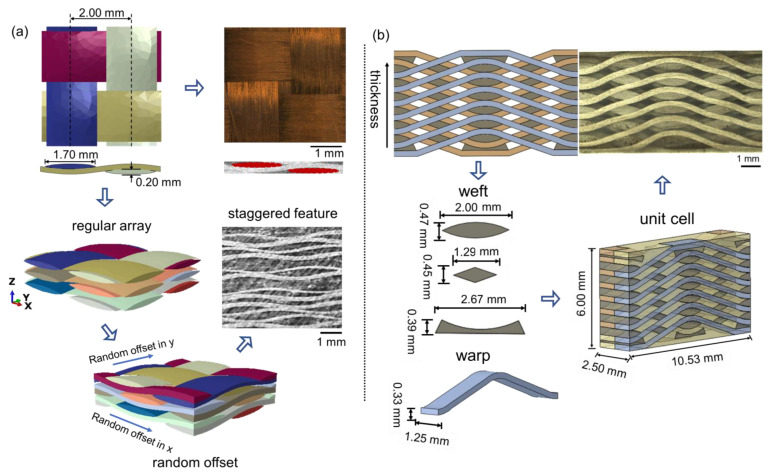
Mesoscale geometry models: (**a**) 2-D PWC; (**b**) 2.5-D AWC.

**Figure 3 polymers-14-03454-f003:**
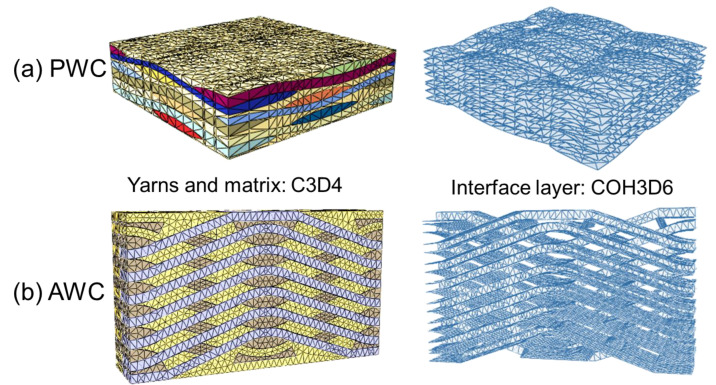
Finite element models: (**a**) 2-D PWC; (**b**) 2.5-D AWC.

**Figure 4 polymers-14-03454-f004:**
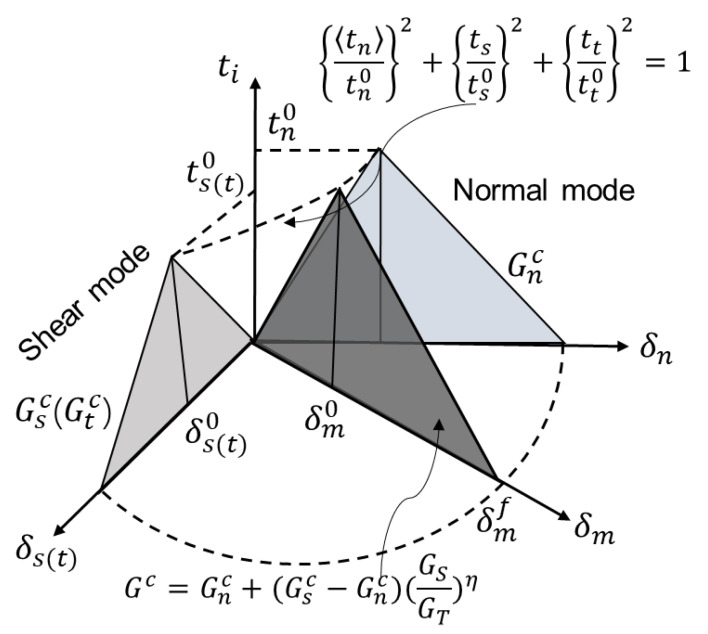
Bilinear traction–separation constitutive model.

**Figure 5 polymers-14-03454-f005:**
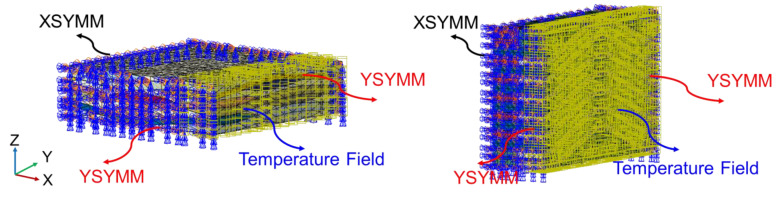
Boundary conditions for 2-D PWC and 2.5-D AWC.

**Figure 6 polymers-14-03454-f006:**
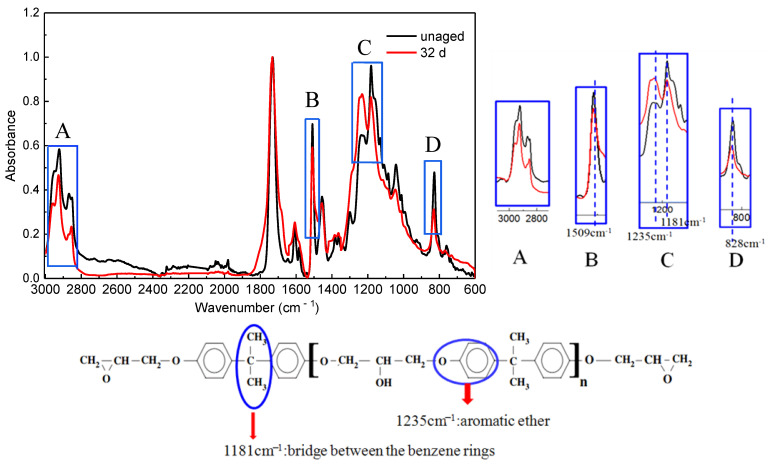
IR spectra of the un-aged resin and the surface of an epoxy cube after ageing at 180 °C for 32 days.

**Figure 7 polymers-14-03454-f007:**
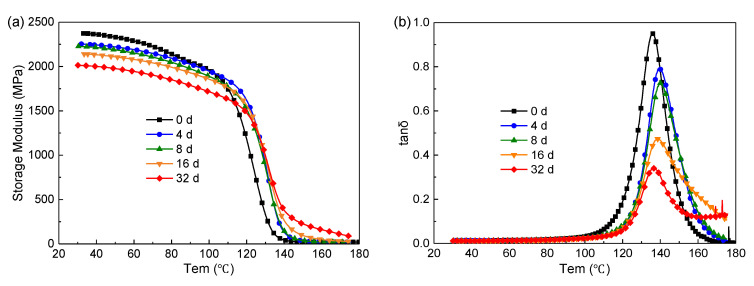
Dynamic thermomechanical behaviors. (**a**) storage modulus; (**b**) loss factor.

**Figure 8 polymers-14-03454-f008:**
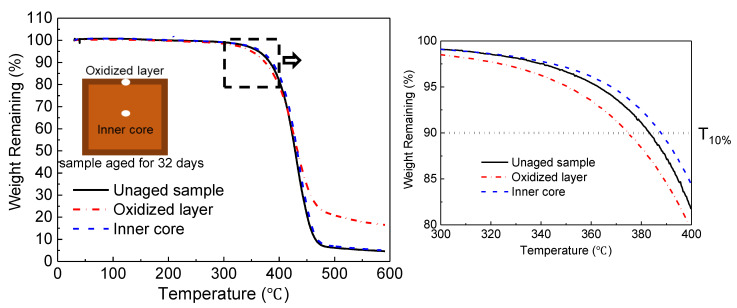
Thermogravimetric analysis.

**Figure 9 polymers-14-03454-f009:**
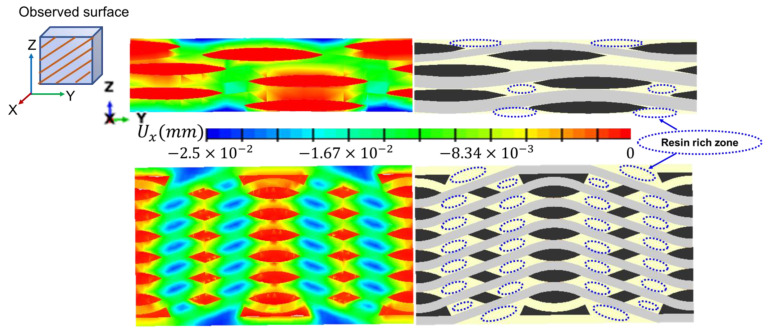
Local shrinkage displacement in PWC and AWC after thermal oxidative ageing for 16 days.

**Figure 10 polymers-14-03454-f010:**
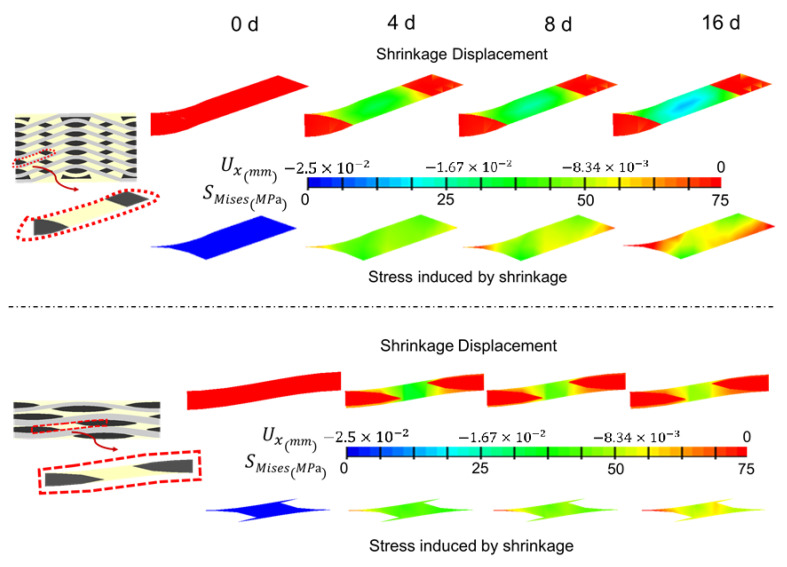
Local shrinkage and internal stress in PWC and AWC with different ageing time.

**Figure 11 polymers-14-03454-f011:**
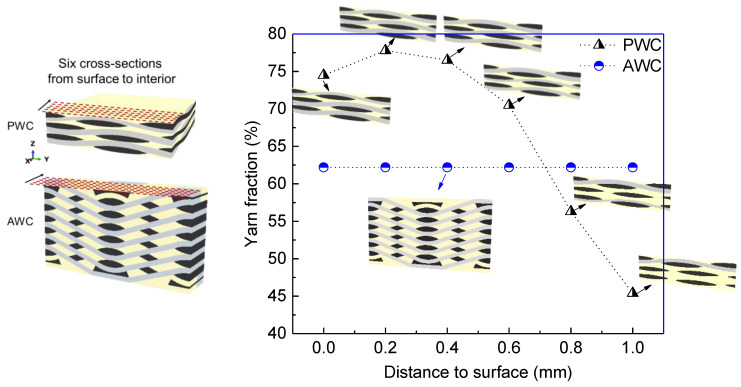
Variations of yarn volume fraction in six cross sections form the surface to interior.

**Figure 12 polymers-14-03454-f012:**
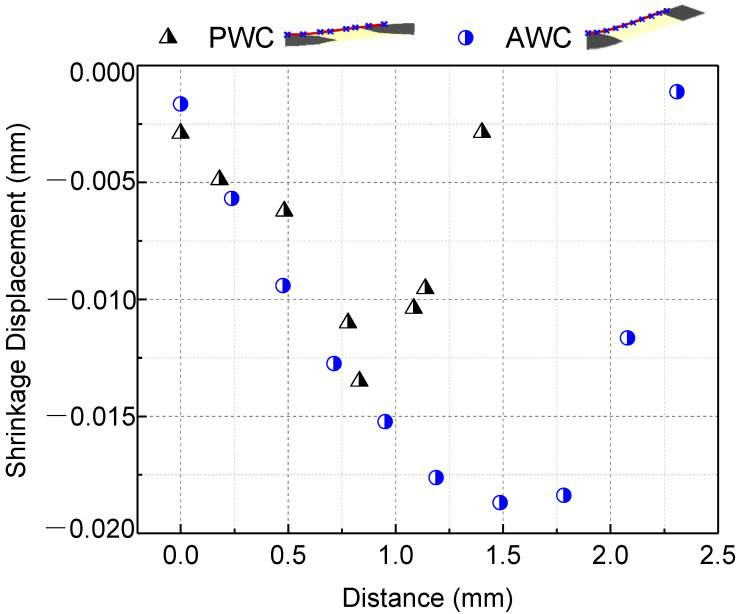
Comparison of shrinkage displacement with yarn distance in PWC and AWC.

**Figure 13 polymers-14-03454-f013:**
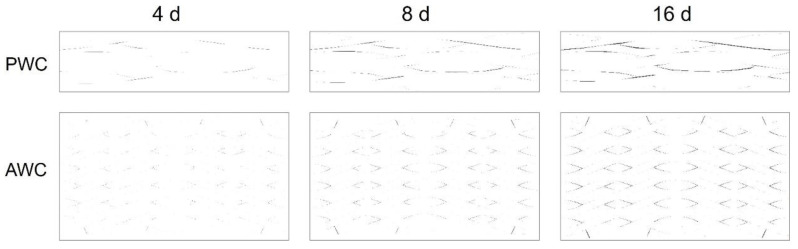
Evolution of interface cracks with the increase of ageing time.

**Figure 14 polymers-14-03454-f014:**
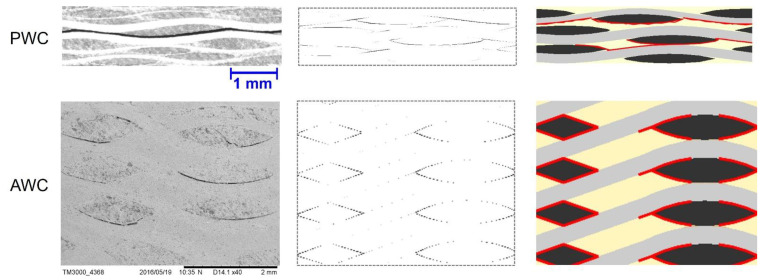
Distribution of interface cracks in AWC and PWC.

**Figure 15 polymers-14-03454-f015:**
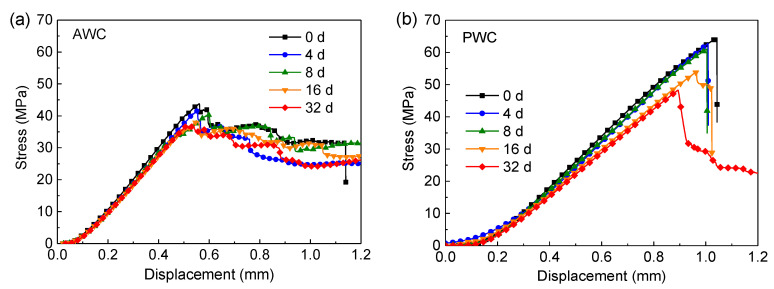
Load–displacement curves under the interlaminar shear tests: (**a**) AWC; (**b**) PWC.

**Figure 16 polymers-14-03454-f016:**
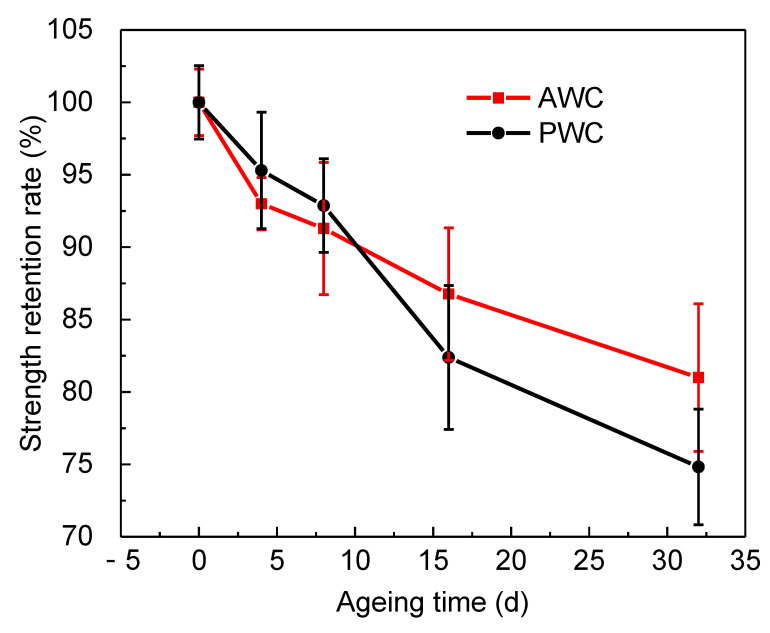
Retention rate of the short beam strength along with ageing time.

**Figure 17 polymers-14-03454-f017:**
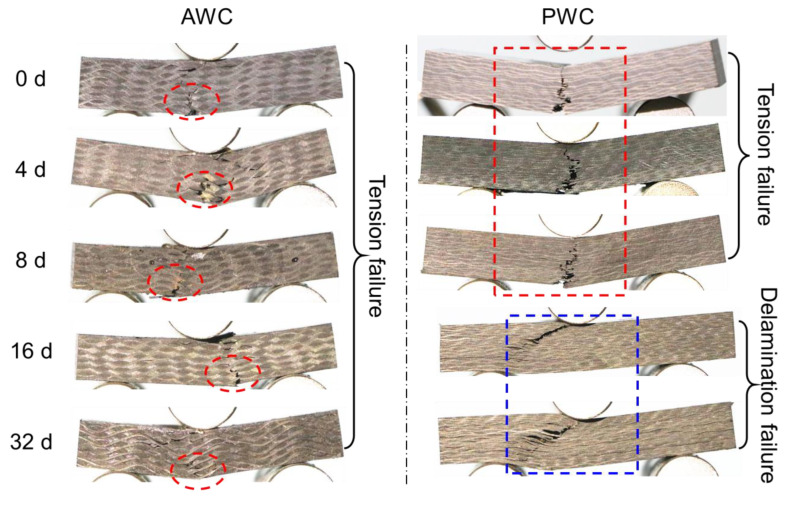
Failure morphologies of AWC and PWC with different ageing time.

**Table 1 polymers-14-03454-t001:** Specifications of the epoxy matrix.

Name	Chemical Component	Viscosity at Room Temperature (MPa·s)	Epoxide Number (eq/100 g)	Blending Ratio
JC-02A	Bisphenol A epoxy resin	1000–3000	0.5–0.53	100:80
JC-02B	Modified anhydride	30–50	-	

**Table 2 polymers-14-03454-t002:** Specifications of the carbon fiber.

Name	Density (g/cm^3^)	Diameter (um)	Carbon Content
Carbon fiber	1.8 ± 0.02	7	≥95%

**Table 3 polymers-14-03454-t003:** Manufacturer Specifications.

Preform	Parameter	Warp	Weft
2-D Plain woven	fiber type	T300-3K	T300-3K
	density/(ends·cm^−1^)	5.0	5.0
	layers	20 (in 0° direction)	
	thickness/mm	0.35 (single layer)	
2.5-D Angle-interlock woven	fiber type	T300-6K	T700-12K
	density/(ends·cm^−1^) *	8.0	3.8
	layers	7	6/8
	thickness/mm	6.20	

* The density/(ends·cm^−1^) means the number of the yarns per centimeter.

**Table 4 polymers-14-03454-t004:** Basic properties of constituents and yarns.

	Carbon Fiber	Epoxy Resin	Fiber Tows
E_11_ (GPa)	230	2.4	142.8
E_22_ = E_33_ (GPa)	14		6.4
G_12_ = G_13_ (GPa)	9	0.89	3.0
*G_23_* (GPa)	5		2.3
ν_12_ = ν_13_	0.25	0.35	0.11
ν_23_	0.3		0.35

**Table 5 polymers-14-03454-t005:** Interfacial parameters for cohesive layers [32].

$K_{n}$ $\left(N/mm^{3}\right)$	$K_{t}\;=\;K_{s}\;$ $\left(N/mm^{3}\right)$	$t_{n}^{0}\;\left(MPa\right)$	$t_{s}^{0}\;=\;t_{t}^{0}$ (MPa)	$G_{n}^{C}$ (N/mm)	$G_{s}^{C}\;=\;G_{t}^{C}\;\left(N/mm\right)$	** *β* **
4 × 10^6^	1 × 10^6^	120	150	0.25	1.0	1.0

## Data Availability

The data presented in this study are available on request from the corresponding author.

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
