# Peer review of "Reinforced Structure Effect on Thermo-Oxidative Stability of Polymer-Matrix Composites: 2-D Plain Woven Composites and 2.5-D Angle-Interlock Woven Composites"

_polymers, 2022, doi:10.3390/polym14173454_

Round 1

Reviewer 1 Report

GENERAL COMMENTS

# In this study the thermo-oxidative stability of carbon fiber polymer matrix composites with different integral reinforced structures was investigated experimentally and numerically. (Specimens of the 2-D plain and 2.5-D angular interlock fabric composites). The question is: Does thermal oxidative aging cause degradation of the matrix and the fiber/matrix interface?

# Interesting subject. Relevant research.

 #The research methodology raises some doubts that I would like to see clarified:

-- Is the degradation because of temperature or because of oxidation (Because of the presence of oxygen)?

-- If the tests were performed for temperatures below the Tg of the resin would the behavior be the same?

SPECIFIC COMMENTS

2 Experimental

Page 2:

-- How did you obtain this volume fraction? What is the procedure (or standard)?

-- What equipment did you use (make, model, location where it was used)?

-- Are these volume fraction or mass fraction values?

Table 1:

In: Table 1. Specifications of preforms.

Replace with:    Table 1. Manufacturer Specifications: (name of manufacturer)

in: Table 1:

Clarify the parameter "density/(ends·cm-1)". For mass density values (mass divided by volume), use International Metric Units. Density ρ, [kg/m3]

4. Results and discussion

Page 11 and page 12:

Figures 13 and 14 must be quality improvements

 Figure 15. For a better analysis of the graphs in Figure 15 the axes should have the same range (e.g. Y: 0-70; X: 0-1.2)

 Page 13 :

In Figures 17, define: What are the failure modes of the AWC specimens?

What are the failure modes of AWC specimens? Use the same names: failure in tension and failure in delamination.

5. Conclusions

Page 14: The conclusions should be improved, particularly by quantifying the results obtained.

Reviewer 2 Report

novetly is unclear?

title is misleading and not meaningful?

the manuscript does not lie in the journal fields (polymers)?

more citation of applications and uses of nonwoven and woven textiles should be included in the introduction part:

sensors 14, no. 7 (2014): 11957-11992. Coatings 10, no. 1 (2020): 58. Fibers 6, no. 2 (2018): 34. Zeitschrift für anorganische und allgemeine Chemie 642, no. 13 (2016): 766-772.

Round 2

Reviewer 2 Report

Revised manuscript can be accepted for publication.

Author Response

Thanks for your suggestions.